# Spheres of Influence: Insights into *Salmonella* Pathogenesis from Intestinal Organoids

**DOI:** 10.3390/microorganisms8040504

**Published:** 2020-04-01

**Authors:** Smriti Verma, Stefania Senger, Bobby J. Cherayil, Christina S. Faherty

**Affiliations:** 1Mucosal Immunology and Biology Research Center, Division of Pediatric Gastroenterology and Nutrition, Massachusetts General Hospital, Charlestown Navy Yard, Boston, 02129 MA, USA; ssenger@mgh.harvard.edu (S.S.); cherayil@helix.mgh.harvard.edu (B.J.C.); csfaherty@mgh.harvard.edu (C.S.F.); 2Harvard Medical School, Boston, 02115 MA, USA

**Keywords:** organoids, enteroids, *Salmonella*, host-pathogen interactions, model systems, infectious diseases, organotypic culture system

## Abstract

The molecular complexity of host-pathogen interactions remains poorly understood in many infectious diseases, particularly in humans due to the limited availability of reliable and specific experimental models. To bridge the gap between classical two-dimensional culture systems, which often involve transformed cell lines that may not have all the physiologic properties of primary cells, and in vivo animal studies, researchers have developed the organoid model system. Organoids are complex three-dimensional structures that are generated in vitro from primary cells and can recapitulate key in vivo properties of an organ such as structural organization, multicellularity, and function. In this review, we discuss how organoids have been deployed in exploring *Salmonella* infection in mice and humans. In addition, we summarize the recent advancements that hold promise to elevate our understanding of the interactions and crosstalk between multiple cell types and the microbiota with *Salmonella*. These models have the potential for improving clinical outcomes and future prophylactic and therapeutic intervention strategies.

## 1. Introduction

Many basic life processes are conserved in all organisms, with similarities in molecular mechanisms often increasing as the phylogenetic distances decrease. This conservation enables investigation into various biological mechanisms and diseases, with the expectation that the knowledge gained from one system will provide insight into the workings of related organisms. The similarity between different organisms is particularly important for studying human biology and diseases, since human experimentation is typically unfeasible or unethical. However, several factors must be considered before employing an organism or derived cells as a model for humans, including ease of maintenance and reproducibility, amenability to genetic and molecular manipulations, and perhaps most importantly, similarity to the characteristics of the human biological process under investigation.

The mainstays of research for many decades have been the cell culture and animal model systems. Classically, the in vitro cell culture methodology involves either growing cell lines that are derived from tumors or are genetically transformed to allow continuous propagation, or culturing primary cells derived from the tissues of humans or experimental animals. Use of cell lines has a significant limitation in that the cells do not recapitulate “normal” cellular physiology. Primary cells closely represent the tissue of origin but are limited in availability, have a finite life-span in vitro without being transformed, and display variability arising from differences in donors. Conventional cell culture comprises growing cells (cell lines/primary cells) either in liquid suspension or as adherent, two dimensional (2D) monolayers on an impermeable solid surface. The cell culture model is also a reductionist approach since the model cannot account for dynamic and complex interactions that occur in vivo, particularly those that result from the interactions with multiple cell types. Although processes such as immune responses, cell signaling, or crosstalk with microbiota have been studied using cell culture models of various cell types either alone or in combination, the biggest disadvantage of this system is that the cells may not fully recapitulate in vivo phenotypes and behaviors. More complex model systems, such as *Caenorhabditis elegans*, *Danio rerio*, *Drosophila melanogaster*, mice, or primates provide a whole organism perspective and allow compensation for some of the limitations of cultured cells as the complexity of the models increase. However, these animal models resemble, but do not completely represent, all the conditions encountered in humans, an issue that is especially important in the understanding of disease and development [1]. Recently, many researchers have developed sophisticated, yet tractable, non-cancerous tissue culture models that bridge the gap between simple 2D cell culture and complex in vivo experiments. Such models are broadly known as organotypic culture systems, which can be composed of various types as detailed below. In general, these models consist of a three-dimensional (3D) organ-like structure that is made up of organ-specific differentiated cells of multiple lineages and that recapitulates the unique organizational and functional characteristics of the corresponding organ in vivo.

Historically, the term “organoid” referred to short-term in vitro cultures of tissues such as lung and intestine [2], or 3D cell aggregates such as spheroids. More recently, it has been applied to organized 3D structures generated in vitro from cells with a whole range of origins, such as tissue segments [3,4] and their derived adult stem cells (ASCs) [5,6], transformed cell lines [7], and pluripotent stem cells (PSCs) [8]. However, as research has progressed, the term “organoid” has taken on a more restricted meaning. Organoids have the following characteristics: (1) self-organization: individual cells arrange in vitro into a 3D structure that mimics the in vivo organ or tissue, (2) multicellularity: organoids are composed of multiple cell types typically found in the organ or tissue in equivalent proportions, 3) functionality: the organoid structure should be able to execute at least some of the organ- or tissue-specific functions, and 4) sustainability: organoids can propagate indefinitely without requiring transformation by maintaining a pool of progenitor cells. In 2012, the International Stem Cell Consortium [9] set guidelines for the nomenclature to be used to define the 3D structures generated in vitro depending on their origin and cellular composition. When referring to intestinal organotypic models, 3D structures that are composed of just epithelial cell types are generally known as “enteroids” if derived from the small intestinal epithelium, or “colonoids” if derived from colonic origin. The term “organoid” is usually reserved for 3D structures containing more than one cell lineage. However, it should be noted that these guidelines have not been uniformly adopted by the field. Many researchers commonly use “organoids” as a blanket term for 3D structures derived from ASCs, PSCs, or comprising transformed cell lines that resemble in vivo 3D architecture and physiology.

Methodologies have been adapted to develop organoids from various normal and cancerous mouse and human tissues including colon [6,10], stomach [11], liver [12,13,14], pancreas [15], and kidney [16] to name a few. These techniques have been well reviewed elsewhere [11,17,18,19,20,21]. For the purposes of this review, we will provide a historical context and examples of the different types of intestinal organoid models (Figure 1), followed by a discussion of the application of the models to the study of *Salmonella* pathogenesis.

## 2. Historical Background

The organoid model systems have been established on the basis of several studies that have contributed to various aspects of the technology [22]. One key advance was the development of cell culture methods that permitted cells to form 3D structures. Many of the non-physiological properties of cells in the conventional 2D cell cultures are believed to be due to the loss of 3D geometry [23]. Several techniques have been developed to deal with this problem, such as the hanging drop method [24], 3D micro-molds [25], and rotating wall vessel (RWV) bioreactors [26], a more advanced method that permits larger-scale production of 3D assemblies. Nickerson, CA et al. (2001) [26] utilized a RWV bioreactor developed by NASA that generates low shear and microgravity, allowing cells to remain in suspension to aggregate and grow three dimensionally. The authors were able to establish a 3D culture of Int-407 cells (derived from a normal fetal intestine but later shown to have HeLa cell contamination), thereby promoting cellular differentiation. The resulting 3D aggregates modeled human in vivo differentiation with well-defined cell-to-cell borders, tight junctions, apical-to-basal polarity, and microvilli development. These features were not present when the Int-407 cells were grown as conventional monolayer cultures. Thus, the study helped to demonstrate that 3D organization provides important physical and mechanical cues that facilitate organogenesis. A 3D platform is generally achieved by using matrices such as Matrigel, which consist of proteins typically found in the extracellular matrix (ECM). The ECM is the non-cellular component present within all tissues and organs, and consists mostly of collagen, enzymes, and glycoproteins to provide both structural and biochemical support to surrounding cells. ECMs have been shown to promote the maintenance of structural and functional characteristics of the tissue of origin [27]. The other major events that propelled the development of organoids were advancements in the understanding of stem cell properties and developmental biology of the intestine. The defining characteristics of stem cells—i.e., the capacity for clonal expansion and the ability to differentiate into daughter cells of multiple lineages—have been vital for the development of organoid technology. In fact, with the right cues daughter cells differentiate into multiple cell types and undergo two processes (1) cell sorting out—the formation of discrete domains by different cell types, and (2) spatially restricted lineage commitment, the specialization of function of a cell based on its position in the tissue architecture [28]. These processes result in complex 3D, self-propagating, multicellular, and functional structures [29]. The improved understanding of intestinal development helped define some of these cues. Studies have shown that different molecular gradients converge to create the micro-environment that shapes the intestinal epithelium. Epidermal growth factor (EGF) and Wingless-related integration site protein (Wnt) are highly active in the crypt and are necessary for proliferation, while bone morphogenetic protein (BMP) and Notch signaling pathways active in the villus control cell fate programming [30,31,32].

Another milestone in the field came in 2007 when Barker and colleagues [33] demonstrated in adult mice that intestinal crypt stem cells, which are responsible for proliferation and migration of the epithelial layer towards the villus tip, express the marker leucine-rich repeat containing G-protein coupled receptor 5 (Lgr5) [33]. Shortly thereafter, Ootani et al. [34] developed a long-term culture of primary mouse intestines using air-liquid interface methodology [34]. Fragments of neonatal mouse intestines containing epithelial and mesenchymal cells were cultured to generate cyst-like structures containing all major cell types found in an adult mouse intestine, including Lgr5^+^ stem cells that the authors demonstrated could be modulated by Wnt signaling [34]. The next critical step occurred in 2009, when Sato and colleagues [5] realized the power of these organ-specific stem cells in 3D cultures by isolating intestinal crypts from adult mice. The authors demonstrated that a single Lgr5^+^ cell could form 3D crypt-villus epithelial structures in the presence of appropriate cues, including an environmental signal from a lamin-rich 3D matrix (Matrigel) and a physiological signal in the form of growth factors (the Wnt agonist R-spondin1, EGF, and the BMP-inhibitor Noggin) [5]. The model recapitulated the physiology and organization of the mouse intestine; and when transplanted into mice, successfully reconstituted crypt-villus units [35,36]. Thus, the adult stem cells (ASCs) develop self-organized structures that consist of various types of intestinal epithelial cells. The authors referred to these structures as “organoids”; however, according to the guidelines set by the International Stem Cell Consortium [9], the model should be called “enteroids” since it was derived from the small intestinal epithelial cells.

Regardless of the definition, this work laid the groundwork for the utilization of pluripotent stem cells (PSCs), either derived from embryonic tissue or generated by inducing stemness in mature somatic cells (iPSCs) [37,38,39,40]. The latter offer the additional advantages that invasive procedures to isolate intestinal or colonic biopsies are not necessary and also iPSCs bypass ethical concerns regarding procurement of embryonic stem cells. The PSC-derived structures consist of epithelial cells surrounded by mesenchymal stromal cells and are referred to as “organoids”. During embryonic development, the mesenchyme induces and specifies epithelial identity and differentiation, and is critical for the maintenance of tissue identity [41,42]. However, for interrogation of epithelial cell-specific functions, the mesenchymal cells may serve as a hindrance. Mithal and colleagues [43] have recently identified a directed differentiation protocol that generates mesenchyme-free human intestinal organoids (HIOs), which were employed to measure cystic fibrosis transmembrane conductance regulator (CFTR) gene function using cystic fibrosis patient-derived iPSC lines. The epithelial cells of organoids are not fully mature and resemble fetal epithelial cells when transcriptionally profiled [44]. The maturation of these organoids requires implantation in vivo, such as on the kidney capsule of immunocompromised mice [8,39,44,45]. The fetal-like nature of PSC-derived organoids presents an advantage as it allows for studies that would otherwise be difficult to conduct given the limited availability and ethical concerns surrounding the use of human fetal tissues. PSC-derived organoids follow a differentiation path observed during development [46] and take one to two months to develop, while tissue-specific ASCs develop organoids by a process that recapitulates tissue repair and is complete within weeks [20]. The epithelial cells derived from ASC (enteroids) represent adult-like mature epithelium. The cells of the enteroids also preserve characteristics of the tissue of origin including a diseased phenotype if derived from patients bearing an intestinal condition. These characteristics make enteroids from ASC a faithful model for intestinal diseases [47,48,49].

## 3. Intestinal Organoids/Enteroids

Intestinal enteroids derived from the intestinal stem cells develop multi-lobed structures (Figure 2) consisting of crypt-villus units, with a clearly defined proliferative, crypt-like zone and a large, non-proliferative differentiated zone surrounding proliferative areas [50]. To generate organoids, the pluripotent stem cells are first programmed towards endoderm development to form the epithelial tube that eventually gives rise to the foregut, midgut, and hindgut in response to a combination of anteriorizing and posteriorizing growth factors provided. Additional mesenchymal cells are also present and consist of myofibroblasts and smooth muscle cells surrounding the epithelial cells, presumably derived from remnant cells of mesodermal lineage after endoderm induction [20,51]. The epithelial cells reach homeostasis with a proliferative rate matching the rate of shedding of the cells at the edge of the monolayer. All major epithelial cell types are generated, including goblet cells, enteroendocrine cells, Paneth cells, antigen-sampling microfold cells (M cells), and columnar enterocytes that have a brush border of apical microvilli. The cells of the intestinal enteoroids are suitably polarized, and produce and secrete mucus from the apical surface. The cellular composition of the organoids/enteroids can be varied depending on the factors that are supplemented during the process of growth and differentiation [34]. Cell types not previously generated or maintained in traditional cell culture have been produced and/or characterized in organoids/enteroids, including the enteroendocrine cells and M cells [52,53]. In 2017, Haber and colleagues [54] profiled epithelial cells from both the mouse small intestine and crypt-derived enteroids generated from the mouse small intestine, with the aim of identifying novel subtypes and defining genetic signatures. As M cells are a scarce cell type comprising only 10% of the rare follicle associated epithelium, the authors did not detect M cells in the transcriptomes profiled from the mouse small intestines. However, small intestine-derived enteroids treated with growth factor Receptor Activator of Nuclear factor Kappa-Β Ligand (RANKL) to generate M cells, displayed M cell specific signatures [54].

A unique property of the enteroids is that the 3D structure of the tissue is not required for terminal cellular differentiation [55,56]. Thus, organoids/enteroids can be dissociated enzymatically and reseeded onto Transwells to generate short-term 2D monolayer cultures consisting of various differentiated epithelial cell types that can then be used for experimentation [55,56,57,58]. Moon and colleagues [55] established epithelial cell monolayers from cultured intestinal spheroids derived from the colon of mice. The differentiated monolayers displayed a robust transepithelial electrical resistance (TEER, a measure of barrier integrity) and the ability to transcytose secretory immunoglobulin A (sIgA) upon stimulation with microbial products [55]. The same group in another study (vanDussen et al., 2015) [56] adapted the technique to generate polarized 2D monolayers from human enteroids that were used to conduct adherence assays with enteropathogenic and diarrheagenic *Escherichia coli* [56]. The monolayers generated from enteroids maintain apical-to-basal polarity and barrier function while allowing easy access to the apical and basolateral compartments. The monolayers also maintain the genetic and functional phenotypes of the organoid/enteroids of origin [56] and can be manipulated to express a crypt-like or villus-like phenotype by changing media composition [49]. This system lends itself to many standard functional assays designed for 2D cells as well as to high throughput assays that cannot be easily performed on spherical enteroids/organoids, including but not limited to adhesion/invasion assays, transmigration assays, and inflammatory immune responses [49,51,55,56,59]. These monolayers can be cultured for up to three weeks (Stefania Senger, unpublished observations). Recent studies [60,61,62] have developed techniques to generate monolayers that exhibited compartmentalization of proliferative crypt-like domains and differentiated villus-like regions closely resembling in vivo distribution. Liu et al. [60] generated self-renewing 2D monolayers by plating intestinal stem cells on a thin layer (10 µM) of Matrigel coated onto glass sheets, while Wang et al. [63] relied on the presence of collagen hydrogels in tissue culture plates. Liu and colleagues [60] also altered the combination of growth factors added to the medium, such as addition of blebbistatin and removal of EGF to improve the survival and growth of the stem cell population. These models do not provide access to the basolateral side of the monolayers. To overcome this limitation, Altay and colleagues [62] cultured mouse-derived intestinal enteroids on Transwells coated with Matrigel that provided proper mechanical stiffness. The authors also boosted the proliferation of the stem cells by supplementing the culture medium for the basolateral compartment with conditioned medium from intestinal sub-epithelial myofibroblasts. The resulting monolayers possessed all cell types found in vivo as well as an effective barrier function. Thus, the 2D monolayers offer in vivo-like structural and functional characteristics with the convenience of the 2D format and should prove quite useful, particularly in studies interrogating effects on intestinal stem cells, including during *Salmonella* infection [64,65].

Overall, organoids/enteroids have several advantages. The models maintain the in vivo tissue architecture, cellular composition, and region-specific differentiation programming [28] while being genetically stable [66]. Organoids/enteroids are amenable to many of the established techniques of molecular analysis and manipulation, such as CRISPR/Cas9 technology [67], lentivirus transduction [42] single-cell RNA sequencing [53], and mass spectrometry [68] to name a few. Organoids/enteroids have been utilized for improving the basic understanding of tissue homeostasis, organogenesis, and physiological functions [4,69]. The systems have also been used to model diseases [49,51], to study host-pathogen interactions [59,65,70,71] and cancer [72], and to test potential vaccines and drugs [73,74,75]. The organoids/enteroids are also used as new tools for personalized medicine, where patient-derived models can serve as platforms for testing treatment options. For example, Dekker and colleagues [76] utilized intestinal enteroids derived from patients with cystic fibrosis to assess the responsiveness to CFTR-modulating drugs. Several large repositories of organoids/enteroids from multiple patients have been established and can be a source of both healthy and diseased cells globally, providing researchers access to a varied genetic background to test their hypotheses [10]. The models even have the potential to be a source for transplantation [35]. Below, we will highlight the use of organoids/enteroids for studying *Salmonella* pathogenesis.

## 4. *Salmonella enterica*

The genus *Salmonella* is a major foodborne pathogen [77]. It comprises two species: *S. bongori* and *S. enterica.* Almost all *Salmonella* organisms that cause disease in humans and domestic animals are serovars belonging to *S. enterica subspecies enterica*. Broadly, the diseases caused by *Salmonella* in humans are of two kinds: 1) systemic febrile illness termed typhoid/enteric fever, and 2) an acute self-limiting gastroenteritis. The serovars that cause typhoid are referred to as typhoidal *Salmonella* and include *S. enterica subsp. enterica* serovar Typhi and *S. enterica subsp. enterica* serovar Paratyphi A, B, and C. *S.* Typhi causes approximately 76.3% of global enteric fever cases [78]. *S.* Typhi and *S.* Paratyphi are restricted to humans and higher primates, and clinical manifestations of the infection include sustained high fever, abdominal pain, headache, weakness, malaise, and transient diarrhea/constipation. Without appropriate and effective antibiotic therapy, the infection may lead to gastrointestinal bleeding, intestinal perforation, septic shock, and death [79,80]. The Non-Typhoidal *Salmonella* (NTS) serovars cause self-limiting gastroenteritis and include *S. enterica subsp. enterica* serovar Typhimurium and *S. enterica subsp. enterica* serovar Enteridis, which are the most prevalent clinical isolates, according to the World Health Organization (WHO). These pathogens are broader in host range and infect humans and animals such as poultry, cattle, reptiles, and amphibians. Infections with NTS typically involve self-limiting diarrhea, stomach cramps, headache, vomiting, and fever that resolve on their own; however, the infection can be severe in children and the elderly and can sometimes be fatal [81]. NTS serovars have been reported to cause an invasive infection similar to typhoid, particularly in sub-Saharan Africa, predominantly in children and HIV-positive adults, with several co-morbidities such as ongoing or recent malaria infection and malnutrition contributing to higher mortality [82,83,84]. *Salmonella* infections represent a considerable economic burden and public health concern in both developing and developed countries. The Center for Disease Control and Prevention (CDC), estimates that 1.35 million infections by *Salmonella* occur in the United States. NTS causes more than 93 million global infections per year, with 155,000 deaths [85]. A study examining the global burden of diseases, injuries, and risk factors in 2017 reported 14.3 million cases of enteric fever globally, with a case fatality of 0.95% resulting in approximately 135,000 deaths [78]. Children, elderly, and those residing in lower-income countries account for the greatest incidences [86,87].

*Salmonella* is acquired via contaminated food and water. Luminal bacteria invade M cells and absorptive enterocytes via a specialized apparatus called the type three secretion system (T3SS) [88] encoded on the *Salmonella* Pathogenicity Islands (SPIs) [89]. The T3SS injects bacterial proteins into host cells allowing the bacteria to essentially commandeer host cellular processes to induce cytoskeletal rearrangements that engulf the bacteria into specialized vesicles called the *Salmonella* containing vacuoles (SCVs) [90]. This invasion process, with subsequent translocation across the epithelium, is followed by the uptake of the bacteria by macrophages and dendritic cells in the intestinal sub-mucosa, where bacterial proteins interfere with phagolysosomal maturation and allow the bacteria to survive inside the cells [91,92,93]. NTS serovars undergo prolific growth in the intestine, while T3SS effectors induce fluid secretion and promote inflammation. Immune signaling via recognition of pathogen-associated molecular patterns such as lipopolysaccharide (LPS) and flagella also induce a robust inflammatory response, which actually provides *Salmonella* with a growth advantage over resident microflora [81,94,95,96]. The immune response eventually limits *Salmonella* growth; nevertheless, the short-term proliferation is sufficient to ensure propagation. Typhoidal *Salmonella* strains elicit a more attenuated inflammatory response in the intestine, especially in terms of limited neutrophil recruitment [79,97,98]. Bacteria migrate to the mesenteric lymph nodes (MLN) and systemically within the reticuloendothelial cells, and as free bacteria in the blood or lymph, to establish new foci of infection in the liver, spleen, bone marrow, and gallbladder [99]. At these new sites the bacteria replicate and re-enter the intestinal lumen via secretion in bile, promoting shedding of the bacteria to continue the cycle of new infections by contaminated food and water. In 3%–5% of cases, the bacteria can persist for long durations in the gallbladder, which serves as a reservoir of chronic infection [100]. Chronic infection with *Salmonella* has been found to be a risk factor for the development of malignant neoplasms, including gallbladder cancer [101,102] and colorectal cancer [102,103].

The emergence of multi-drug resistance to conventional antibiotics complicates the treatment of *Salmonella* infection [104,105,106,107]. Antibiotic treatment destroys the resident microflora, provides a niche for *Salmonella* to proliferate, and may lead to increased levels of bacterial shedding [108]. Additionally, bacterial populations that express an antibiotic-tolerant phenotype can evade treatment and persist, causing relapses of the infection as well as the evolution of bacterial virulence [109,110]. Asymptomatic carriers act as reservoirs, contribute to the continued propagation of the pathogen, and are particularly important for food safety considerations. Currently, there are no effective vaccines against gastrointestinal *Salmonella* infections. Several typhoid vaccines are available and licensed in many countries; however, robust protection is limited and has been associated with injection site reactions. Furthermore, the vaccines have not been widely adopted by public health programs [111]. With the significant aging populations in both developed and developing countries [112,113], more people are at risk for severe consequences of *Salmonella* infections. Advances in the mechanistic understanding of *Salmonella* infections will facilitate the development of improved control strategies, particularly, safe and effective vaccines. The broad conservation of host responses as well as the molecular machinery used by *Salmonella* strains during infection of various hosts, namely T3SSs encoded by SPI1 and SPI2 that enable invasion of epithelial cells and subsequent intracellular survival, allows several model systems to be applied for *Salmonella* research [114]. Each non-human model has pros and cons, and varies in the ability to recapitulate natural infection. Given the host-specific aspects of infection physiology, it is necessary to be cautious in applying the results from these model systems to human patients. Additionally, with *Salmonella* being an important tool to understand host physiology, metabolism, immune function, and interactions with microbes, human-specific investigations are valuable to the research community.

## 5. Model Systems to Study *Salmonella* Biology

Transformed cell lines such as Cos-1, MDCK, HeLa, HepG2, CaCo2, and T-84 have been used to carry out several fundamental studies on *Salmonella* pathogenicity, such as the identification of the T3SS and the SPIs [88,115]. However, these cells have the drawback noted earlier inthat the cells do not adequately represent the physiological characteristics of normal human tissue [116,117]. Explant tissue cultures have organotypic properties that can be vital for studies on development and physiology, but are limited by culturing difficulties and short life span [118,119]. Classically, animal models have offered solutions for several of these limitations, and have been used to corroborate data obtained from other model systems as well as to investigate the deeper molecular mechanism of infection. For example, *S.* Typhimurium is a natural pathogen of calves and causes a gastroenteritis with clinical and pathological manifestations similar to humans, namely diarrhea, anorexia, fever, localized infection, and neutrophil infiltration [120,121]. Meanwhile, the bovine ligated loop was instrumental in characterizing fluid accumulation and host inflammatory responses following *S.* Typhimurium infection. For example, mutants lacking the invasion proteins SipA, SopA, and SopD were shown to have little to no effect on the ability of *S.* Typhimurium to invade epithelial cells, but were shown to reduce the fluid accumulation and neutrophil immigration in bovine loops [122,123].

More importantly, the vast majority of studies on *Salmonella* pathogenesis have been conducted in the murine model, including studies for the development of new vaccines. This model has been useful in clarifying various aspects of in vivo *Salmonella* pathogenesis; however, it does have limitations with respect to its ability to faithfully reproduce all aspects of *Salmonella* infection in humans. *S.* Typhimurium causes two very different types of diseases in human and mice. In humans, the infection is localized, dominated by the infiltration of neutrophils and self-limiting. In mice, *S.* Typhimurium spreads systemically, with slow infiltration of mononuclear inflammatory cells and little or no localized intestinal tissue injury. The host responses involved are also dramatically different. Since the clinical manifestations and pathology of mice infected with *S.* Typhimurium resemble those observed in humans with *S.* Typhi infection, the model has been used as a surrogate to study typhoid pathogenesis [124]. A mouse model of *S.* Typhimurium-induced enterocolitis has been developed and involves pre-treatment of mice with a single dose of streptomycin. This procedure diminishes the colonization resistance by the commensal microbiota, allowing *Salmonella* to grow to high densities in the cecum and large intestine and trigger acute gastroenteritis [125]. Mouse models have also been developed to mimic chronic infections of *Salmonella* observed in certain carrier individuals, which typically involve infecting susceptible mice with avirulent strains, or sub-lethal doses of *S.* Typhimurium in resistant mice. Experimental interpretations from the mouse model may not translate to human disease. *S.* Typhi and *S.* Typhimurium share about 89% of their genes, with approximately 500 genes unique to *S.* Typhimurium and 600 genes unique to *S.* Typhi, including the genes encoding typhoid toxin and the immunoprotective capsule [126,127]. *S.* Typhi and *S.* Paratyphi also have pseudogenes as well as small sequence differences in genes encoding the T3SS apparatuses and related effectors that may have important implications in pathogenesis. It is possible that virulence factors that may have no role in *S.* Typhi-mediated infection in humans may be important for mouse infection by *S.* Typhimurium and vice versa. Finally, *S.* Typhi genes required for causing typhoid but absent in *S.* Typhimurium cannot be studied easily using the latter. Typhoid fever can be induced experimentally by oral infection in higher primates or human volunteers, but these studies come with their own set of difficulties and ethical objections. Researchers have attempted to compensate for the discrepancies by developing humanized mice, i.e., immunodeficient mice engrafted with human hematopoietic cells [128,129], but these models are cumbersome, expensive, and still do not guarantee that mouse-specific factors will not add complexities or variations to the data generated. In addition, host genetic background has been found to play a role in susceptibility to invasive NTS infections, a concept that borders on the realm of personalized medicine [130]. Thus, organoids offer a promising model to mechanistically study host-specific aspects of infection.

## 6. Organoids/Enteroids in *Salmonella* Biology

One of the earliest studies that utilized organotypic 3D structures to investigate *Salmonella* pathology was carried out by Nickerson, CA et al., in 2001 [26]. As described above, the authors generated 3D organotypic cultures by growing human embryonic intestinal cell line Int-407 in RWV bioreactors and subsequently infected the cells with *S.* Typhimurium. The resulting infection was quite different from what had previously been observed in monolayer cultures. There was minimal loss of structural integrity, lower ability of the bacteria to adhere to and invade epithelia, and lowered expression of cytokine in 3D Int-407 aggregates as compared to infected Int-407 monolayers. Since the authors observed that the 3D Int-407 aggregates more closely resembled in vivo characteristics (tissue organization, tight junctions, apical-to-basal polarity, microvilli development, expression of extracellular and basement membrane proteins, and greater M cell glycosylation pattern), the authors concluded that the infection phenotypes observed in the 3D aggregates were likely representative of an in vivo infection. This study laid the groundwork for the use of 3D organotypic cultures to study *Salmonella* biology. The following section will highlight research performed in both mouse and human models that has improved our understanding of *Salmonella* pathogenesis (Figure 3).

### 6.1. Mouse-Derived Models

Following the establishment of protocols to generate crypt-derived mouse intestinal enteroids (referred to as organoids by the authors) by Sato et al., Zhang and colleagues [65] in 2014 utilized the system to analyze interaction of *S.* Typhimurium with epithelial cells. The authors visualized bacterial infection, while also observing bacterial-induced disruption of tight junctions, activation of the nuclear factor kappa-light-chain-enhancer of activated B cells (NF-kB)-mediated inflammatory response, and a decrease in the stem cell marker Lgr5. The authors noted that these observations were similar to findings in animal models. The caveat to this study is that the *Salmonella* were not delivered into the lumen of the enteroids, the location of the initial contact of bacteria with epithelial cells in vivo; instead, the bacteria were added to the medium and came in contact with the enteroids basolaterally. Nevertheless, this study established mouse-derived enteroids as a model system for studying *Salmonella* infection biology. 

Since this initial study, enteroids have been used to interrogate various aspects of *Salmonella* pathology, including investigating cell types that were previously not accessible to study in vitro. Farin and colleagues [131], in a 2014 study, used mouse intestinal enteroids to study the control of Paneth cell (PC) degranulation in response to bacteria or bacterial molecules such as LPS. The authors found that PC degranulation did not occur upon stimulation with microbial molecules or *Salmonella,* but was induced by a novel mechanism requiring only the presence of recombinant interferon gamma (IFN-γ) [131]. In another study, Wilson and colleagues [132] interrogated the antimicrobial role of Paneth cell α-defensin peptides. The authors developed small intestinal epithelial enteroids from both wild-type mice or mice mutated for α-defensin production (*Mmp7^-/-^* mice, MMP7 is a matrix metalloprotease that is required to generate bactericidally active α-defensins in mice [133]), and infected the enteroids with *S.* Typhimurium by microinjecting the bacteria directly into the lumen. The absence of mature α-defensins reduced the intra-luminal bacterial killing, which could be partially restored by expression of the human defensin HD5 [132]. This study demonstrated the contribution of α-defensins to the innate immune response to *Salmonella,* which previously had been a challenge to examine since most of the earlier experimental systems inadequately recapitulated in vivo cellular processes [134]. 

*Salmonella* has been suggested to contribute to the development of cancer by epidemiological studies [102]. Scanu and colleague [135] probed this phenomenon in a 2015 study using the case of gallbladder cancer (GBC). The authors derived gallbladder enteroids from mice carrying mutations that inactivate p53 and are known to be found in GBC patients in India, where the disease is prevalent. When exposed to wild-type *S.* Typhimurium, single cells derived from the gallbladder enteroids carrying these predisposing mutations generated new enteroids that exhibited growth factor independence, which is one of the hallmarks of transformation, and had histopathological features consistent with neoplastic transformation, thus establishing a direct association between *Salmonella* and cancer. To delve into the mechanism of this transformation, the authors looked at the *Salmonella* T3SS-mediated activation of AKT or mitogen-activated protein (MAP) kinase pathways, which have been shown to be elevated in human cancers. The signals activated by AKT and MAPK were found to be key in driving the cellular transformation and were sustained even after the eradication of the *Salmonella* infection. Studies have shown that AKT and MAPK pathways are activated by other bacteria and viruses that have been associated with various cancers [136,137,138,139]. Although the authors employed a murine gallbladder enteroid model and *S.* Typhimurium to study GBC, the authors proposed that the AKT and MAPK pathways are activated by both *S.* Typhimurium and *S.* Typhi serovars, and contribute to development of cancer that is associated with chronic *S.* Typhi infection in humans. Chronic infection by *Salmonella* has also been found to be a risk factor for developing cancers in the ascending and transverse colon [103]. However, detailed mechanistic studies of *Salmonella*-associated colon carcinogenesis need to be carried out and organoid model systems may prove to be extremely useful for this purpose. 

Finally, another important aspect of *Salmonella* is the interaction of the pathogen with the host microbiome. Lu and colleagues [140] recently demonstrated that *Lactobacillus acidophilus*, a well-established probiotic bacterium, can alleviate damage caused by *S.* Typhimurium. Earlier studies had shown that *L. acidophilus* can inhibit adhesion of *Salmonella* to CaCo2 cells [141]. In this study, the authors extended the mechanism of protection to include the effects on the host. *L. acidophilus* altered the differentiation of epithelial cells in crypt-derived enteroids by impeding the *Salmonella*-mediated expansion of Paneth cells [131], thus maintaining homeostasis and appropriate epithelial composition during the infection. This study not only improved our understanding of the role of *L. acidophilus* in protecting the epithelial lining, but also demonstrated the ability to include microbiota-specific analyses to study *Salmonella* infection with enteroids/organoids.

### 6.2. Human-Derived Models

The potential to gain significant insight into *Salmonella* pathogenesis is particularly relevant in relation to the use of human-specific organoids, especially since these models possess human genetic specificities absent in mice. Studies have interrogated the usefulness of intestinal enteroids/organoids derived from human ASCs or PSCs to understand complex interactions between the epithelium and *Salmonella*. In 2015 Forbester and colleagues [70] used RNA sequencing to examine the epithelial transcriptional signature following injection of *S.* Typhimurium into the lumen of organoids derived from human induced PSCs (hiPSCs). The analysis showed significant up-regulation of genes for cytokine-mediated signaling, NF-κB activation, angiogenesis, and chemotaxis. Enhanced release of pro-inflammatory cytokines IL-8, IL-6, and TNF-α was also confirmed. The findings were consistent with prior studies in animal and mouse organoid models, thus establishing the human organoids as a viable infection model for *Salmonella*. The study also demonstrated that a noninvasive mutant strain (deficient in *invA* gene) could be used in the model to examine *Salmonella* pathogenicity and the functionality of defined mutants [70]. Furthermore, the authors generated an RNA sequencing data set following basolateral administration of *Salmonella* to the organoids. Interestingly, 49 of the 100 most highly upregulated genes were also significantly induced in the data set obtained by microinjecting the bacteria for apical infection. The data provide credence to the results of the Zhang and colleagues study [65], which documented similar patterns of gene expression upon basolateral administration of *Salmonella* to mouse organoids as had been observed earlier in literature with other model systems. Thus, the hiPSC organoids maintain a conserved response to *Salmonella* infection and provide a human-specific model for pathogenesis studies. 

A subsequent study further demonstrated the validity of hiPSCs as a model to study human-specific responses to *Salmonella* infection. Using the same model system, the role of the cytokine IL-22 in priming intestinal epithelial cells towards a more effective response to *S.* Typhimurium was also explored [142]. The study showed that IL-22 pre-treated hiPSC-derived organoids increased phagolysosomal fusion leading to enhanced antimicrobial activity. Thus, this study confirmed earlier observations made in mouse organoids [143]. 

The fidelity of organoid-derived data in representing human disease was further demonstrated in 2018 by Nickerson, KP and colleagues [71], who compared infection of human tissue biopsies and human intestinal enteroid-derived monolayers seeded on a 2D Transwell system, and observed that the enteroid-derived epithelial monolayers recapitulated *S.* Typhi infection observations made in the tissue biopsy model. The authors also carried out transcriptional profiling of both the host tissue and the bacteria in order to determine early critical interactions. Infection with *S.* Typhi significantly down-regulated several host genes, including those involved in activation of the mucosal immune response, bacterial clearance, and cytoskeletal rearrangement. Interestingly in this model, a down regulation of SPI1 genes in *S.* Typhi was observed. This work demonstrated that *S.* Typhi reduces intestinal inflammation by limiting the induction of pathogen-induced processes through the regulation of virulence gene expression, which is a characteristic feature of human infection with *S.* Typhi. Transmission electron microscopic comparisons of the tissues and human organoid-derived epithelial monolayers showed that the monolayers reproduced the cytoskeletal arrangements, microvilli destruction, and vesicle-bound bacteria observed in tissues. There were no changes observed in paracellular permeability, increased death of host cells, or bacterial association with M cells, suggesting divergence from *S.* Typhimurium infection in mice. This study highlights the ability of organoids to compare human-specific responses to each *Salmonella* serovar, which is important in the context of translational capacity for developing prophylactic or therapeutic intervention strategies against *S.* Typhimurium versus *S.* Typhi infections. 

Despite the multiple advantages of the organoids as an experimental system, the technology is still in its infancy and has certain limitations. The complex structure of the organoids poses a practical limitation in accessing the internal luminal compartment. Researchers have used microinjections to access the apical epithelium. This approach may preserve the internal microenvironment, but is resource-intensive, may not allow synchronous exposure and suffers from variability in volume that can be injected due to heterogeneity of the organoid/enteroid sizes. In addition, the lumen of 3D organoid/enteroid accumulates cellular debris, which may bind bacteria or hamper interactions with the apical membranes. As noted above (in Section 3), researchers have turned to organoid/enteroid-derived 2D monolayers to better access the apical side of the model and enable more efficient, user-friendly analyses in a multiple-well plate format. However, this modification can limit the number of processes that can be interrogated, especially when considering the lack of 3D structure. Interestingly, Co and colleagues [144] demonstrated in a recent study that the polarity of human enteroids could be reversed such that the apical surface faced the medium and was readily accessible. The enteroids released mucus and extruded cells outwards into the culture medium rather than having the cells embedded in the basement membrane. Using enteroids with reversed polarity, the authors showed that *S.* Typhimurium invades and induces actin ruffles more efficiently at the apical surface compared to the basolateral surface. The authors observed a more diffuse process of epithelial invasion rather than invasion only or predominantly at the M cells [144], which confirmed the *S.* Typhi observations by Nickerson, KP et al. [71].

Current organoid/enteroid models are devoid of muscles, innervation, vascularization, and immune cells. There are a couple of approaches being carried out to increase the complexity of organoid models, including co-culturing techniques. In 2011, Salerno-Goncalves and colleagues [7] generated an organotypic model using the human ileococal adenocarcinoma cell line HCT-8 and adding primary endothelial cells, fibroblasts, and peripheral blood mononuclear cells (PBMCs), which they used in a 2019 study to probe the crosstalk between these cell types during infection with *S.* Typhi, *S.* Paratyphi A, or *S.* Paratyphi B [145]. An ECM composed of collagen-I enriched with other gastrointestinal basement proteins was embedded with the fibroblasts and epithelial cells, and transferred to a RWV bioreactor containing epithelial cells. Under low microgravity and low shear conditions, the HCT-8 cells behaved as multi-potent progenitor cells and gave rise to multiple cell types, including absorptive enterocytes, goblet cells, and M cells. After one to two weeks, PBMCs were added to the system. The co-culture model was then infected with the various *Salmonella* serovars to compare responses to the three strains. The authors found that the presence of the immune cells in the model resulted in secretion of the cytokines IL-1β and CCL3, while secretion of cytokines IL-6 and TNF-α was enhanced. Using depletion experiments, the authors showed that macrophages were the PBMC cell type responsible for the enhanced secretion of IL-6 and TNF-α. The authors further used the Transwell system to show that supernatants from organotypic models built with whole or macrophage-depleted PBMCs infected with the three *Salmonella* strains varied in their ability to elicit transmigration of macrophages and neutrophils [145]. Interestingly, the two immune cells displayed crosstalk during infections with *S*. Paratyphi A and *S*. Paratyphi B, such that the presence of macrophages in the co-culture reduced neutrophil migration as compared to the system built without macrophages [145]. This study illustrates that co-cultures can aid in probing the contribution of immune cells to *Salmonella* infection at the mucosal surface. Finally, this model has also been used to assess the inflammatory response to several candidate *S.* Typhi vaccine strains in comparison to the response elicited by the oral vaccine strain Ty21a strain and its parent wild-type Ty2 stain [146]. Salerno-Goncalves and colleagues [146] found that specific changes to the genetic makeup of the candidate vaccine strains (in the form of deletions of specific metabolic genes) elicited host changes in intestinal permeability, inflammatory cytokine secretion, as well as activation of innate immunity pathways. Higgins and colleagues [73] also used the model to test the inflammatory response of an *S.* Typhimurium vaccine strain that they generated. These studies highlight the usefulness of co-cultured organoid/enteroid models in assessing important factors to be considered while designing vaccines.

Schulte and colleagues [147] generated a co-culture system of human intestinal epithelial cell line (Caco-2), primary human microvascular endothelial cells, primary intestinal collagen scaffold, and PBMCs in a Transwell set up. Using GFP-labeled *S.* Typhimurium, microscopy, and flow cytometry, the authors demonstrated that the bacteria can be found in epithelial but not endothelial cells, thus modeling the epithelium-restricted infection of humans with *S.* Typhimurium. These findings are in contrast to those of Spadoni and colleagues [99] in the mouse model of *S.* Typhimurium infections where a breach of the gut-vascular barrier by the bacteria was observed. The endothelial cells respond to the infection process by bringing about changes in transcription of various genes and releasing the phagocyte chemoattractant IL-8. Such models, ideally with enteroid/organoid-derived cells replacing cell lines where used, should prove to be extremely useful and versatile in interrogating the role of different immune cells, vasculature, and the related crosstalk with epithelial cells during infection with *Salmonella,* especially for *S.* Typhi, where the bacteria spread systemically both as free bacteria and within reticuloendothelial cells [148].

## 7. Future Directions

Currently, researchers have little or no control over how cells self-organize into organoids. The physical environment of the enteroids/organoids in the form of the 3D scaffold provides cues such as adhesive ligands and stiffness. The current ECMs are derived from animals, are poorly defined, may show batch to batch variation that can lead to heterogeneous growth and differentiation between the organoids generated at different times, and are not mechanically pliable after plating. Thus, improvements to the 3D scaffolds are expected to yield better consistency, which would facilitate mechanistic infection studies and even provide a better platform for clinical applications. Some recent work [149] has focused on using chemically defined 3D scaffolds to improve the uniformity of the environmental cues for growth and differentiation. Indeed, a better defined and mechanically dynamic matrix would increase the potential of organoid technologies for therapeutic development (also reviewed in [150] and [151]).

Another important requirement is the further characterization of organoids/enteroids to ensure that the models faithfully represent the in vivo human physiology so that in vitro analyses with the system will be relevant for subsequent clinical development. Recent work in brain organoids has shown that important differences exist in the expression profiles between cells derived from organoids as compared to human brain cells [130]. Similar studies must be carried out in intestinal organoid/enteroid models, and indeed in all organoid models, to ensure fidelity of representation. 

The enteric nervous system (ENS) carries out important functions in the gastrointestinal tract, such as motility and contractility, regulation of blood flow, maintenance of epithelial barrier, and fluid exchange [7]. Future investigations of the role of the ENS in *Salmonella* infection will be critical to our mechanistic understanding of the pathogen. Recent studies have shown that the neurons in the gut also protect against *S.* Typhimurium infections [152,153]. For example, Lai and colleagues [153] demonstrated that the pain-sensing neurons that lie beneath Peyer’s patches in the gastrointestinal tracts of mice, become activated and release the neuropeptide calcitonin gene-related peptide (CGRP) upon detecting *S.* Typhimurium. This process results in a decreased number of M cells while increasing the levels of segmented filamentous bacteria (SFB) that can protect against *Salmonella* infection [153]. In light of studies like this, it becomes essential to probe and understand the interaction of *Salmonella* with the ENS. To facilitate analyses with the ENS, researchers have developed an ENS-HIO model. Workman and colleagues [154] generated vagal-like neural crest cells (NCCs, the precursors of ENS) from PSCs and incorporated them into human intestinal organoids via direct co-culture. The NCCs subsequently differentiated into multiple cell types including neurons and glia. The authors demonstrated that the neurons were functional by observing a response to oscillations in calcium. The ENS-HIO was then transplanted into mice and displayed contractile activity. Given the recent findings of the role of neurons in potential protection against *Salmonella* infection, this model has tremendous promise to further our understanding of the dynamics of the ENS-*Salmonella* interaction. 

Methods to incorporate the microbiome into organoid-based models are essential to future *Salmonella* research. It is well known that *Salmonella* must successfully compete with the resident microflora in order to infect the intestinal epithelium, and several competitive mechanisms deployed by *Salmonella* have been elucidated [155]. Most often, microbiome studies are carried out in germ-free or humanized mice; however, there are significant differences between the composition of the microbiota and host physiology when using these animals as a model of humans. The healthy gut contains several anaerobic bacteria in the lumen given the decreasing oxygen gradient that extends from the anaerobic lumen to the hypoxic epithelium. Despite the complexities of microbiota interactions and oxygen levels, a few studies have made progress in incorporating the microbiota into organoid models or reproducing anaerobic environments such as the 2020 study by Lu et al., described above [140]. Karve and colleagues [156] in a 2017 study incubated induced human intestinal organoids (iHIOs) with human neutrophils to model innate cellular responses to commensal and pathogenic *Escherichia coli* in which neutrophil recruitment was monitored by microscopy. The authors were able to culture commensal *E. coli*, despite the fact that the bacterial strain is a facultative anaerobe. Greater recruitment of the neutrophils occurred when the organoids had been injected with pathogenic bacteria as compared to saline or commensal bacteria. In another study by Leslie and colleagues [157], the authors demonstrated that anaerobic *Clostridium difficile* could remain viable for at least 12h when injected into the lumen of HIOs, indicating the reproducibility of the appropriate oxygen levels in the model. Finally, LeBlay and colleagues [158] have devised a specialized bioreactor for cultivation of human intestinal microbiota that will enable researchers to grow a wide range of commensal bacteria for analyses. The authors immobilized fecal microbiota from a two year old child in gel beads and cultured under anaerobic conditions with continuous flow of medium containing chyme. The bacteria grew as biofilms and formed stable populations. *S.* Typhimurium also immobilized onto beads was added to the reactor to simulate gut infection in children. This step was followed by addition of two concentrations of amoxicillin, and the effects of the treatment on the microbial composition and the metabolites generated were analyzed. The authors observed a strong disturbance in the microbial composition upon antibiotic treatment, with *Bifidobateria* significantly decreasing in numbers while *C. cocoides*-*E.rectales* group strongly increased. S. Typhimurium levels were also strongly decreased upon amoxicillin treatment, but returned to previous levels upon interruption of the antibiotic treatment. Antibiotic treatment also resulted in a decrease in the concentration of the metabolites acetate and butyrate, which remained at lower levels even on the withdrawal of antibiotic treatment. Indeed, these studies show that the incorporation of the microbiota into human-specific organoid systems will allow mechanistic studies investigating the crosstalk between host physiology, microbiota, and *Salmonella* pathogenicity. 

## 8. Conclusions

Infections caused by *S. enterica* remain a major health concern world-wide. Models used to study the disease pathology so far have provided valuable advancements. However, there remains a disconnect between what works at the bench versus at the bedside, particularly in the case of vaccines. The development of organoids/enteroids offers a tremendous opportunity to bridge this gap by bringing human-specific factors into the research models (Figure 3). 

Gastrointestinal organoid and enteroid models have been shown to capture the cellularity, organization, and complexity of the intestine in vivo along with providing the flexibility of an in vitro system. These models have provided new and fundamental knowledge in human physiology, pathology and the molecular basis of host-microbe interactions. In addition, recent studies have seen efforts to build upon existing paradigms. Research is being conducted to improve the fidelity and reproducibility of the organoid model systems. Co-culture systems have been developed allowing the integration of various cell types, which allows us to better understand and interrogate the crosstalk between a wide-range of potential combinations of lineages in vivo. Most importantly, the integration of immune cell types is likely to help in understanding how *Salmonella* is viewed by and responds to immune processes, particularly at the mucosal epithelium that is the first site of contact and thus an important stage of the infection process for the development of vaccines. There have also been studies that have integrated the microbiota into the organoid model and future work may delve into the role of an individual’s microbiota in *Salmonella* pathogenesis. Despite its potential, it is important to keep in mind certain limitations of organoid-based systems, the most significant being that the models have to be thoroughly characterized for their ability to represent in vivo conditions for appropriate translation into the clinic. The addition of multiple cell types into organoid co-cultures only recapitulates a part of the body i.e., the organ from which they are derived. The results of organoid co-cultures have to be complemented with whole organism studies and compared to human clinical findings.

We anticipate that the organoid and enteroid models will play key roles in future advancements in the understanding of *Salmonella* pathogenesis. Their use will facilitate research in drug development, host-microbe interactions, crosstalk between the host, microbiota and pathogens, and personalized medicine, and will contribute towards the development of successful vaccines for *Salmonella* Typhimurium and *Salmonella* Typhi.

## Figures and Tables

**Figure 1 microorganisms-08-00504-f001:**
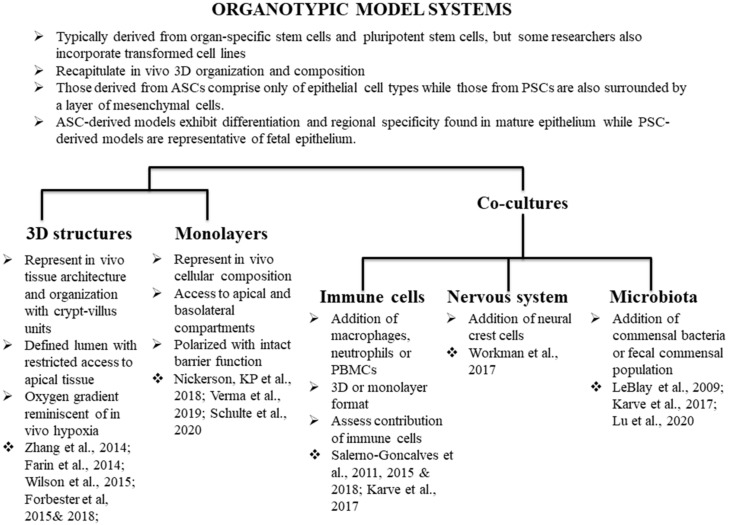
Use of organotypic models in *Salmonella* biology. We highlight studies that have utilized organotypic models to better understand *Salmonella* biology, as well as studies that do not directly pertain to *Salmonella* research but have the potential to be deployed for better understanding of the dynamics of host-*Salmonella* interactions.

**Figure 2 microorganisms-08-00504-f002:**
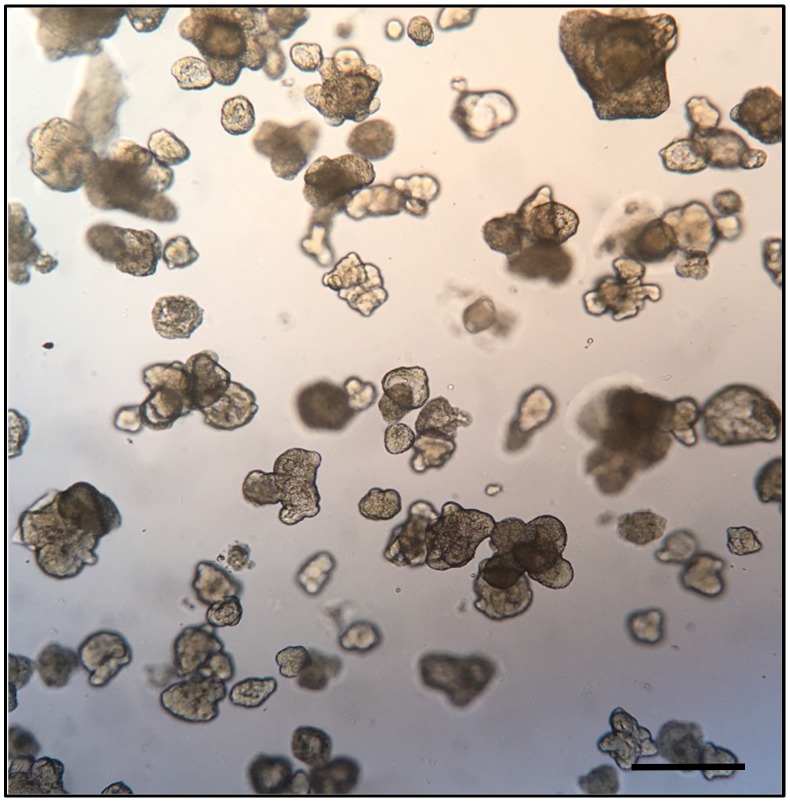
Bright field microphotograph depicting a representative field of human duodenum-derived enteroids in Matrigel. Each of the structures in the figure represents an enteroid at 8 days of culture, consisting of a 3D cellular aggregate organized into an epithelial monolayer resembling that of the small intestine. The apical surface of the monolayer faces the center of the enteroid while the basolateral surface faces the exterior. Bar scale indicates 1.0 mm; image from Stefania Senger, unpublished data.

**Figure 3 microorganisms-08-00504-f003:**
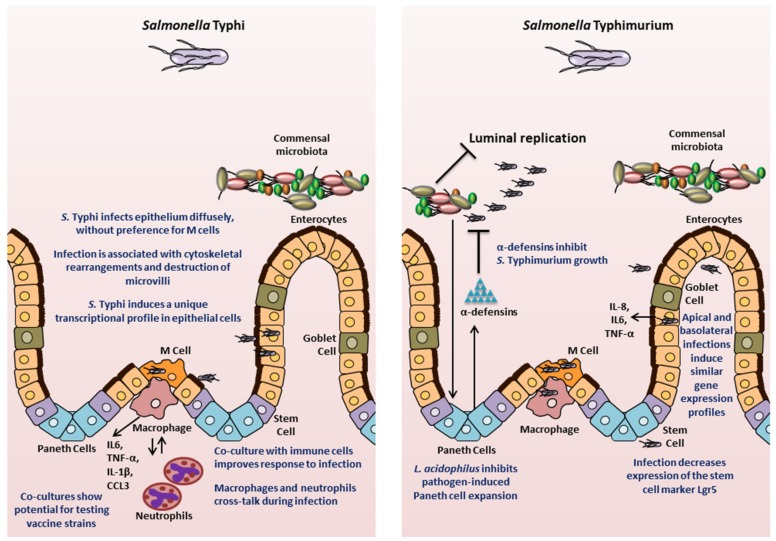
Insights into *Salmonella* pathogenesis from intestinal organoids/enteroids. The key findings for *S.* Typhi and *S.* Typhimurium are highlighted.

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
