# Peer review of "Spheres of Influence: Insights into Salmonella Pathogenesis from Intestinal Organoids"

_microorganisms, 2020, doi:10.3390/microorganisms8040504_

Round 1

Reviewer 1 Report

In this piece, Verma et al review the current literature on organoid-based studies of Salmonella pathogenesis. The review is thoroughly referenced and covers the topic in a balanced fashion. The authors write with clarity and have structured the manuscript for easy reading. The manuscript suffers somewhat from that the number of papers published on the specific topic (i.e. Salmonella infections in organoids) is still limited. This has given rise to extensive chapters also covering Salmonella pathogenesis and organoid approaches in general. However, in the opinion of this reviewer, such a broad scope may be warranted, considering that these are early days for the research field.

In summary, I have some minor points that the authors should adjust or clarify before publication can be recommended.

Specific comments with line numbers:

  1. Line 12, Abstract: Recommend to change “complexities…remain” à ”complexity…remains”
  2. Lines 14-15, Abstract: Organoids do not only bridge the gap between 2D and 3D, they also bridge the gap between the transformed cell lines that have been used before, and primary cells which better maintain physiological signalling. This sentence should be adjusted to clarify that aspect.
  3. Line 37: Cell culture could mean anything. I advise to change to “…has been the cell line culture model”. Similar clarifications would also be warranted for the following sentences (lines 38-42).
  4. Lines 52-54: The authors here present organotypic culture like we would be talking about one single model system. In fact, this term encompasses a wide variety of experimental models, which are detailed later in the manuscript. Adjust this sentence accordingly.
  5. Lines 68-70: This nomenclature praxis (e.g. enteroids, colonoids) applies specifically to gut organotypic models, which should be clarified here. Furthermore, it should be noted that this 2012 nomenclature has not been globally adopted by the field, which may warrant some minor clarifications in Figure 1 (see below).
  6. Adjustments to Figure 1:

6a. At the top, the heading in the left column should read “ENTEROIDS / COLONOIDS”. Since the word “ORGANOIDS” in the mind of many readers has a much broader meaning than the 2012 nomenclature paper suggests, I would also propose to change, or clarify the header for the right column.

6b. The references in the figure are distinct from the characteristics indicated by the box symbols. Some other symbol than empty boxes should therefore be used for indicating the references.

6c. Many of the references indicated in Figure 1 do not match with the claim that this figure describes the “use of organoids/enteroids in Salmonella biology”. Either those references should be removed, or the figure and figure legend adjusted. For example, The Schulte et al 2011 reference is not work done in the typical enteroid, colonoid, or organoid models described in this figure, but rather in a cell line-based co-culture system. The Noel et al 2017 reference cannot be found in the reference list, the Karve et al 2017 reference deals with E.coli infection, and the Le Blay et al 2009 study has nothing to do with organoids/enteroids.

  1. Line 24: Please check that ref 24 is really correct. I do not have access to this reference, but seems surprising to cite a paper from 1907.
  2. Line 176: “organoid” à ”organoids”
  3. Line 220: “the model maintains” à “the models maintain”
  4. Lines 265-272: In addition to M-cells, Salmonella also frequently invade into absorptive epithelial cells in many different host species. This should be acknowledged here.
  5. Lines 326-327: I would advise to change the sentence “This model, however, has some glaring deficiencies”. This statement is true only if all one cares about is human disease. Since many Salmonella serovars are broad host spectrum pathogens, studies of the infection process in mice is still highly relevant for understanding fundamental principles. What is clear is that NTS Salmonella infection progresses differently in humans and laboratory mice.
  6. Adjustments to Figure 2:

12a. The authors are advised to use the same sentence structure for all comments added into this figure. For example, “L. acidophilus inhibits Salmonella-…” and “Decrease in the stem cell marker…” do not follow the same sentence structure, which gives a somewhat arbitrary impression.

12b. Also, this figure could use some polishing of the item design, e.g. consistency in shape and colouring choice. For instance, the M-cells and macrophages look hand drawn in a rush, and harmonize poorly with the symmetrical outlines of other cell type symbols.

Author Response

My coauthors and I are grateful to the reviewers for their overall positive response to our manuscript and for their helpful suggestions for improving it further. We have addressed each of their comments as detailed below and have made corresponding changes to the manuscript. The changes have been highlighted in yellow in the marked up version of the revised text.

Reviewer 1

Line 12, Abstract: Recommend to change “complexities…remain” à ”complexity…remains”

We have followed the reviewer’s suggestion and have modified the sentence (Line 11).

Lines 14-15, Abstract: Organoids do not only bridge the gap between 2D and 3D, they also bridge the gap between the transformed cell lines that have been used before, and primary cells which better maintain physiological signalling. This sentence should be adjusted to clarify that aspect.

We have followed the reviewer’s suggestion and have modified the sentence (Line 13-15).

Line 37: Cell culture could mean anything. I advise to change to “…has been the cell line culture model”. Similar clarifications would also be warranted for the following sentences (lines 38-42).

We understand the reviewer’s concern and have modified the relevant sentences (Line 38-44).

Lines 52-54: The authors here present organotypic culture like we would be talking about one single model system. In fact, this term encompasses a wide variety of experimental models, which are detailed later in the manuscript. Adjust this sentence accordingly.

We thank the reviewer for pointing this out. We have modified the sentence structure to clarify this point (Line 57-60).

Lines 68-70: This nomenclature praxis (e.g. enteroids, colonoids) applies specifically to gut organotypic models, which should be clarified here. Furthermore, it should be noted that this 2012 nomenclature has not been globally adopted by the field, which may warrant some minor clarifications in Figure 1 (see below).

We thank the reviewer for pointing out the discrepancies in the figure. We have made appropriate modifications. We have also made changes to the text regarding the nomenclature used (or not) for enteroids/organoids (Line 77-78 and Line 80-83).

Adjustments to Figure 1:

6a. At the top, the heading in the left column should read “ENTEROIDS / COLONOIDS”. Since the word “ORGANOIDS” in the mind of many readers has a much broader meaning than the 2012 nomenclature paper suggests, I would also propose to change, or clarify the header for the right column.

We have heeded the reviewer’s concern and changed the title to “Organotypic model systems”.

6b. The references in the figure are distinct from the characteristics indicated by the box symbols. Some other symbol than empty boxes should therefore be used for indicating the references.

We now use different bullet signs for the characteristics and the reference list.

6c. Many of the references indicated in Figure 1 do not match with the claim that this figure describes the “use of organoids/enteroids in Salmonella biology”. Either those references should be removed, or the figure and figure legend adjusted. For example, The Schulte et al 2011 reference is not work done in the typical enteroid, colonoid, or organoid models described in this figure, but rather in a cell line-based co-culture system. The Noel et al 2017 reference cannot be found in the reference list, the Karve et al 2017 reference deals with E.coli infection, and the Le Blay et al 2009 study has nothing to do with organoids/enteroids.

We have altered the references mentioned. We also mention in the legend that studies carried out to study Salmonella and those that have the potential to be used for Salmonella have been highlighted (Line92-95).

Line 24: Please check that ref 24 is really correct. I do not have access to this reference, but seems surprising to cite a paper from 1907.

We have updated the reference to a more recent one. Kelm JM, Fussenegger M. 2004. Microscale tissue engineering using gravity-enforced cell assembly. Trends Biotechnol 22:195–202.

Line 176: “organoid” à ”organoids”

We have followed the reviewer’s suggestion and have made the change (Line 189).

Line 220: “the model maintains” à “the models maintain”

We have followed the reviewer’s suggestion and have made the change (Line 248).

Lines 265-272: In addition to M-cells, Salmonella also frequently invade into absorptive epithelial cells in many different host species. This should be acknowledged here.

We have followed the reviewer’s suggestion and have modified the sentence (Line 294-295).

Lines 326-327: I would advise to change the sentence “This model, however, has some glaring deficiencies”. This statement is true only if all one cares about is human disease. Since many Salmonella serovars are broad host spectrum pathogens, studies of the infection process in mice is still highly relevant for understanding fundamental principles. What is clear is that NTS Salmonella infection progresses differently in humans and laboratory mice.

We understand the reviewer’s concern and have modified the relevant sentence (Line 360-362).  

Adjustments to Figure 2:

12a. The authors are advised to use the same sentence structure for all comments added into this figure. For example, “L. acidophilus inhibits Salmonella-…” and “Decrease in the stem cell marker…” do not follow the same sentence structure, which gives a somewhat arbitrary impression.

We have modified sentence structures as suggested by the reviewer.

12b. Also, this figure could use some polishing of the item design, e.g. consistency in shape and colouring choice. For instance, the M-cells and macrophages look hand drawn in a rush, and harmonize poorly with the symmetrical outlines of other cell type symbols.

We thank the reviewer for the suggestions and have incorporated them into the figure, which is now Figure 3.

Reviewer 2 Report

This review by Verma et al. discusses new advances in the development of enteric organoid models, and their use in exploring pathogenesis and bacteria-host interactions during Salmonella infection.  The authors give an overview of the history of tissue organoid models, and compare the merits and drawbacks of these models with those of classical in vitro and animal-based experiments.  They then proceed to describe a variety of studies that used mouse- and human-derived organoid systems to study various aspects of Salmonella infection, with an emphasis on new innovations in the creation of enteric organoids that enabled more accurate recapitulation of in vivo infection, which in turn led to new insights into Salmonella pathogenesis and the host response. They end with a thoughtful and well-balanced discussion of the drawbacks that yet remain with these methods.  Overall, this is a very clear and well-written article.   Major concerns: none.   Minor concerns: 1) if possible, representative images/pictures comparing 2D models and the organoids would be nice 2)line 239: “sorovars” should be serovars” 3)focused on salmonella infection in the case of gallbladder cancer using enteroids which seems to be outside the scope of this review: are there intestinal cancers that salmonella affects? 4)Be careful about the wording in lines 192 and 193 in regards to reference 55 (Moon et al. 2014: the intestinal spheroids are not capable of producing sIgA (only B cells can do this). Microbial products were used to stimulate intestinal spheroids to increase pIgR expression (receptor required for sIgA transcytosis) leading to increase sIgA transcytosis.  5)Lines 284-285, “the bacteria can persist for long durations in the gallbladder, which serves as a reservoir of infection that is at increased risk of developing cancer” is confusingly worded – does the presence of bacteria in the gallbladder raise the risk of cancer in the gallbladder specifically? Or does it raise the risk of cancer in the intestine by prolonging the infection? 6)Lines 364-365, “These observations led the authors to conclude that the 3D culture model was more reflective of in vivo infection”: a sentence or two reminding the reader of the features of in vivo infection that are being compared here would be useful 7)Line 391, “enteriods” should be “enteroids”.

Author Response

My coauthors and I are grateful to the reviewers for their overall positive response to our manuscript and for their helpful suggestions for improving it further. We have addressed each of their comments as detailed below and have made corresponding changes to the manuscript. The changes have been highlighted in yellow in the marked up version of the revised text.

Reviewer 2

1) if possible, representative images/pictures comparing 2D models and the organoids would be nice

We thank the reviewer for the suggestion and have included an image of human duodenum-derived enteroids 8 days post culture. This image is now Figure 2.

2) line 239: “sorovars” should be serovars”

The typographical error has been corrected (Line 268).

3) focused on salmonella infection in the case of gallbladder cancer using enteroids which seems to be outside the scope of this review: are there intestinal cancers that salmonella affects?

The gallbladder is considered a part of the digestive system and Salmonella Typhi carriage is a significant risk factor for developing gallbladder cancer. Therefore we thought it prudent to discuss the study by Scanu et al. Salmonella infections have also been implicated as risk factors for the development of colorectal cancers, pancreatic cancers and lung cancers by epidemiological studies, which we have now mentioned in the text (Line 452-455). Detailed mechanistic analyses are awaited. Organoids could prove to be an important model system to study these cancers.

4)Be careful about the wording in lines 192 and 193 in regards to reference 55 (Moon et al. 2014: the intestinal spheroids are not capable of producing sIgA (only B cells can do this). Microbial products were used to stimulate intestinal spheroids to increase pIgR expression (receptor required for sIgA transcytosis) leading to increase sIgA transcytosis. 

We thank the reviewer for pointing out this oversight. We have made appropriate changes. Line 220-221.

5)Lines 284-285, “the bacteria can persist for long durations in the gallbladder, which serves as a reservoir of infection that is at increased risk of developing cancer” is confusingly worded – does the presence of bacteria in the gallbladder raise the risk of cancer in the gallbladder specifically? Or does it raise the risk of cancer in the intestine by prolonging the infection?

We have clarified the statement (Line 315-317).

6)Lines 364-365, “These observations led the authors to conclude that the 3D culture model was more reflective of in vivo infection”: a sentence or two reminding the reader of the features of in vivo infection that are being compared here would be useful

We have clarified the statement (Line 399-403).

7)Line 391, “enteriods” should be “enteroids”.

The typographical error has been corrected (Line 429).
